# Meropenem plus Ertapenem and Ceftazidime–Avibactam plus Aztreonam for the Treatment of Ventilator Associated Pneumonia Caused by Pan-Drug Resistant *Klebsiella pneumonia*

**DOI:** 10.3390/antibiotics13020141

**Published:** 2024-01-31

**Authors:** Konstantinos Mantzarlis, Efstratios Manoulakas, Kyriaki Parisi, Evaggelia Sdroulia, Nikolaos Zapaniotis, Vassiliki Tsolaki, Epaminondas Zakynthinos, Demosthenes Makris

**Affiliations:** 1Department of Critical Care, University Hospital of Larissa, School of Medicine, University of Thessaly, 41110 Thessaly, Greece; emanoulakas@uth.gr (E.M.); parisi@uth.gr (K.P.); buchbox1973@gmail.com (E.S.); vastsolaki@uth.gr (V.T.); ezakynth@uth.gr (E.Z.); dimomakris@med.uth.gr (D.M.); 2Department of Microbiology, University Hospital of Larissa, School of Medicine, University of Thessaly, 41110 Thessaly, Greece; nzapaniotis@yahoo.com

**Keywords:** pan-drug resistant *K. pneumonia*, ceftazidime–avibactam + aztreonam treatment, double carbapenem therapy, ventilator associated pneumonia (VAP), survival, mechanical ventilation, Sequential Organ Failure Assessment (SOFA) score, Clinical Pulmonary Infection Score (CPIS)

## Abstract

Introduction: Gram-negative bacteria (GNB) account for about 70% of infections in the intensive care unit (ICU) setting and are associated with significant morbidity and mortality. In recent years, pan-drug resistant (PDR) strains, strains that are not susceptible to any antibiotic, have been emerged and new treatment strategies are required. Results: Fifty eligible patients were recruited in the three groups. A statistically significant reduction in the Sequential Organ Failure Assessment (SOFA) score was observed in the control group on day 4 in comparison to day 0 of VAP (*p* = 0.005). The Clinical Pulmonary Infection Score (CPIS) was also reduced on day 4 (*p* = 0.0016) and day 7 in comparison to day 0 (*p* = 0.001). Patients that received combination therapy, CAZ–AVI + ATM and DCT, presented with a lower SOFA score and CPIS on day 7 in comparison to day 0 (*p* = 0.0288 and *p* = 0.037, respectively). No differences in the ΔSOFA score and ΔCPIS were found between the groups. The control group presented with a significantly lower ICU stay and duration of mechanical ventilation (*p* = 0.03 and *p* = 0.02, respectively). There was no difference in mortality. Materials and methods: This is a retrospective analysis. This study was conducted in a mixed ICU in the University Hospital of Larissa, Thessaly, Greece during a three-year period (2020-2022). Patients suffering from ventilator associated pneumonia (VAP) due to carbapenem-resistant *K. pneumonia* (CR-KP) were divided in three different groups: the first one was treated using ceftazidime–avibactam plus aztreonam (CAZ–AVI + ATM group), the second was treated using double carbapenems (DCT group), and the last one (control group) received appropriate therapy since the strain was susceptible in vitro to at least to one antibiotic. Conclusions: Treatment with CAZ–AVI +ATM or DCT may offer a clinical benefit in patients suffering with infections due to PDR *K. pneumoniae*. Larger studies are required to confirm our findings.

## 1. Introduction

Gram-negative bacteria (GNB) account for a considerable percentage of intensive care unit (ICU) infections. They are associated with significant morbidity and mortality [1]. Therefore, susceptibility to available antibiotics is a major concern for physicians. At the same time, resistance to antibiotics has been growing in recent years, and has become an emerging concern in critical care [2]. Carbapenemases confer the largest antibiotic resistance spectrum, since they can hydrolyze β-lactamases. In recent years, carbapenem resistant *Klebsiella pneumoniae* strains (CR-KP) and, moreover, pan-drug resistant (PDR) strains (strains that are not susceptible to any antibiotic) have emerged. Infections caused by such strains constitute a major problem since limited therapeutic options are available, the level of treatment failure is high, and the mortality rate is therefore increased [3,4,5]. Thus, antibiotic combinations could be considered in the absence of other therapeutic options.

In vitro data support the use of combination of aztreonam (ATM) with ceftazidime–avibactam (CAZ–AVI) [6,7]. When the combination of CAZ–AVI is insufficient to counter pathogen’s mechanisms of resistance, a third antibiotic, ATM, is added to restore susceptibility. Different methods were used to examine the synergistic effects of the combination, such as the disk diffusion method and the E-test, and the results were promising. On the other hand, clinical data are limited. An alternative antibiotic strategy for PDR strains is double-carbapenem therapy (DCT) that was previously first attempted in Greece [8], demonstrated a bactericidal effect, and had clinical success. The rationale for such treatment is that ertapenem may serve as a suicide inhibitor to bolster high concentrations of the second carbapenem due to its high affinity for carbapenemases. On this basis, DCT may lead to microbiological success [9]. An alternative basis for this treatment is the reduction in the bacterial load achieved initially by ertapenem that will increase the activity of the second carbapenem. However, clinical data about DCT, especially its effectiveness and safety, are also very limited.

The aim of our study was to compare the effectiveness of treatment with CAZ–AVI + ATM or DCT on Sequential Organ Failure Assessment (SOFA) score and Clinical Pulmonary Infection Score (CPIS) decline over the first days of the treatment of patients with ventilator associated pneumonia (VAP) due to PDR *K. pneumoniae*. CPIS lacks sensitivity and specificity and its use alone is not recommended by ATS/IDSA guidelines for VAP to initiate or discontinue antibiotic treatment, but its use is simple and, furthermore, is the only semiobjective assessment of several clinical factors indicative of pneumonia, according to the same guidelines [10].

VAP was defined according to Centers for Disease Control and Prevention (CDC) criteria (www.cdc.gov/nhsn/pdfs/pscmanual/10-vae_final.pdf, accessed on 15 December 2022). The administration of antibiotics that displayed documented in vitro susceptibility according to the breakpoints established by the European Committee on Antimicrobial Susceptibility Testing (EUCAST) [11] was considered appropriate therapy. As a PDR defined the strain of *K. pneumoniae* that was not susceptible to any agents in any antimicrobial categories (i.e., bacterial isolates not susceptible to any clinically available drug) [12].

## 2. Results

The study was conducted over a three-year period between January 2020 and December 2022. A total of 903 patients were studied. One hundred and two (11.3%) patients with a positive for CR-KP culture were identified. Twenty-four patients (2.7%) were excluded because CR-KP was considered to constitute colonization, and thus inclusion criterion (c) was not met. The rest, seventy-eight patients (8.6%), presented with VAP. Among them, twenty-three (2.5%) met exclusion criterion (c) and five cases (0.6%) met exclusion criterion (b) and were subsequently excluded. The rest, fifty patients (5.6%) were divided into three different groups: twelve patients were recruited into the CAZ–AVI +ATM group (1.3%), eleven patients (1.2%) were recruited into the DCT group, and twenty-seven patients (3%) were recruited into the control group. Table 1 represents the characteristics of the patients during ICU admission (baseline characteristics). Mechanical ventilation duration before VAP diagnosis was 9.8 (2.2) days for the control group, 7.7 (0.4) days for the DCT group, and 8.2 (0.4) days for the CAZ–AVI + ATM group (*p* = 0.69). Furthermore, patients received antibiotic treatment for 7.13 (1.5), 6.17 (0.4), and 6.5 (0.5) days, respectively (*p* = 0.86). There was no statistically significant difference in baseline characteristics.

All patients received empirical antibiotic treatment after VAP diagnosis. The initial antibiotic treatment was modified after culture results. For the control group, empirical treatment was appropriate for 18 patients. The rest received antibiotics active against the pathogen 1.6 (0.7) days after VAP diagnosis, since the empirical treatment was not appropriate. The time elapsed from the VAP diagnosis to meropenem and ertapenem administration for the DCT group, and ceftazidime–avibactam and aztreonam for the CAZ–AVI + ATM group was 1.9 (0.8) and 2.1 (0.9) days, respectively (*p* = 0.80). None of the patients in the two groups received combination therapy as an initial empirical treatment. For the control group, colistin was the active agent for twenty-two patients, whereas seven of them also received tigecycline. The other five patients from the control group received aminoglycosides, and especially gentamicin as an active treatment. In addition, three of them received also tigecycline. The CAZ–AVI + ATM group received treatment for VAP for 9.5 (0.6) days, the DCT group for 9.4 (0.5) days, and the control group for 9.7 (0.6) days (*p* = 0.64).

The control group presented with a statistically higher SOFA score in comparison to the other two groups on day 0 of VAP [5.7 (0.7) for the CAZ–AVI +ATM group vs. 6 (0.8) for the DCT group vs. 8.5 (0.7) for the control group, *p* = 0.02]. On day 4, the SOFA score was also higher [4.7 (0.6) vs. 5.5 (0.9) vs. 7.3 (0.7), respectively, *p* = 0.047] (Table 2). For the control group, the SOFA score declined on day 4 in comparison to day 0 of VAP [8.5 (0.7) vs. 7.3 (0.7), *p* = 0.005] (Table 2). Patients that received CAZ–AVI + ATM and DCT presented with a statistically significant decline in the SOFA score between day 0 and day 7 of VAP [5.8 (0.5) vs. 4.6 (0.5), respectively, *p* = 0.029]. For each combination group separately, the SOFA score declined but not significantly for both day 4 and day 7 of VAP in comparison to day 0 (Table 2). There was no statistically significant difference in ΔSOFA score between the groups for both day 4 and day 7 of VAP.

The CPIS on day 0 of VAP was higher for the control group in comparison to the other two groups [6.3 (0.5) for the CAZ–AVI +ATM group vs. 5.6 (0.5) for the DCT group vs. 7.3 (0.3) for the control group, *p* = 0.02] (Table 3). Only on the control group was the CPIS significantly lower on day 4 and day 7 in comparison to day 0 of VAP [7.3 (0.3) vs. 5.7 (0.4), *p* = 0.0016, and 7.3 (0.3) vs. 5.6 (0.4), *p* = 0.001, respectively]. The other two groups did not present with a statistically significant decrease in CPIS for both day 4 and day 7 of VAP in comparison to day 0, but for all the patients that received CAZ–AVI + ATM and DCT, the CPIS was significantly lower on day 7 in comparison to day 0 of VAP [5.9 (0.4) vs. 5.0 (0.3), *p* = 0.037]. There was no statistically significant difference in ΔCPIS for day 4 and day 7 between the groups.

The durations of mechanical ventilation and ICU stay were significantly lower for the control group in comparison to the other groups [70.7 (25.3) for the CAZ–AVI +ATM group vs. 44.4 (6.3) for the DCT group vs. 36.4 (9.3) for the control group, *p* = 0.02, and 75 (33.4) vs. 56.5 (8.4) vs. 39.1 (9.7), *p* = 0.03, respectively]. Mortality did not differ between the three groups (Table 4).

## 3. Discussion

In the present study, we aimed to compare the outcome of patients suffering from VAP that were treated using two different antibiotic combinations: CAZ–AVI + ATM and DCT, and more specifically, meropenem and ertapenem. This is the first study that considers critically ill mechanically ventilated patients. Our findings suggest that the combination treatments led to SOFA score and CPIS decline, even if the decline was not similar to that in patients that received appropriate therapy for VAP, since they suffered from CR-KP that was sensitive to at least one antibiotic. Moreover, mechanical ventilation and ICU stay were shorter for the control group, but there was no difference in mortality.

Appropriate therapy seems to be more effective. The control group patients presented with better clinical outcomes compared to the other two groups. The abovementioned affirmation is elicited by the fact that there was a statistically significant reduction in the CPIS during the course of VAP. In addition, the SOFA score was significantly lower for control group patients, despite a higher severity of the illness being observed on day 0 of VAP, since the control group patients presented with higher SOFA score and CPIS in comparison to both combination therapy groups. The fact that the control group eventually had a better clinical outcome suggests the effectiveness of treatment. It is important to notice that clinical improvement concerns the improvement of both the symptoms of pneumonia (evaluated with the CPIS) and the level of multi-organ failure secondary to sepsis syndrome (evaluated with the SOFA score). Additional data that potentiate the clinical improvement thesis for the patients that received the appropriate therapy is that the combination therapy groups had longer durations of mechanical ventilation and ICU stay.

On the other hand, the patients that received combination therapy had lower SOFA scores and CPIS a week after the initiation of treatment, and also ΔSOFA and ΔCPIS score for days 4 and day 7 of VAP presented no differences between the three groups. Moreover, mortality was not statistically different. Therefore, we can conclude that combination therapy may not be as beneficial as appropriate therapy. On the other hand, the decrease in the CPIS and SOFA score suggests that combination therapy can affect the course of the infection.

Clinical data about the effectiveness of the combination of CAZ–AVI + ATM for the treatment of infections due to carbapenem resistant strains are lacking. Only case reports and case series have been published [13,14,15,16,17,18]. The largest study so far was conducted in three hospitals in Italy and Greece [19]. According to the authors, the CAZ–AVI + ATM combination offered a therapeutic advantage compared to other active antibiotics, such as colistin, aminoglycosides, and tigecycline. Our results are in part in agreement to the previous study, but it should be noted that the *K. pneumonia* strains in our study did not present susceptibility to any antibiotic, including CAZ–AVI and ATM. Moreover, the participants were suffering from a different type of infection (VAP instead of blood stream infection), and all of them were mechanically ventilated in contrast to the previous study where only one third of patients was under mechanical ventilation. In our study, mechanical ventilation, ICU stay, and mortality were higher in the CAZ–AVI + ATM group in comparison to DCT group, but the differences were not statistically significant. On the other hand, the CAZ–AVI + ATM group presented at baseline with a lower PaO_2_/FiO_2_ ratio, but again, this difference was not statistically significant in comparison to the DCT group.

There are scarce clinical data about the efficacy of DCT for the treatment of infections due to PDR *K. pneumoniae*. Most of the studies are case reports with a limited number of patients and without control groups [20,21,22,23,24,25,26]. The types of included infections were mostly blood stream infections and pneumonia. In most cases, the patients presented clinical improvement, microbiological clearance, and no relapse during the follow-up period. On the other hand, clinical studies comparing DCT to other antibiotic combinations are rare. In the largest study published to date [27], patients with documented CR-KP infection who were treated using DCT were matched based on the severity of the illness with those who were treated using standard therapy. The group that received DCT had more favorable outcomes, and especially lower mortality. In two other studies [28,29], DCT therapy was also effective compared mostly to colistin, even if mortality was not statistically different between the groups. In a systematic review and meta-analysis of studies where DCT therapy was used [30], the authors concluded that patients that received DCT presented with a similar efficacy response and lower mortality, and thus DCT could be used as an alternative therapeutic option. But they highlighted the need for more high-quality clinical trials to address the efficacy and safety of the specific antibiotic combination. Differences between the results of the abovementioned studies and the results of our study may have many explanations: in our study, the participants presented a specific type of infection, due to a specific pathogen, and were mechanically ventilated and strictly hospitalized in the ICU where many other factors are involved in mortality. Furthermore, the strains in the DCT group did not present susceptibility to any other antibiotic regimen. It must be underlined that in our study, in concordance with previous studies, patients that received DCT therapy also presented with improvement.

This study has limitations. It was performed in a single center and therefore the results should be interpreted cautiously. The number of participants was relatively small, a fact that may limit the generalizability of the results. CPIS sensitivity and specificity for VAP is relatively low. The mechanisms of resistance were not studied for each isolate. For susceptibility testing, no other methods than Vitek 2 and disk diffusion were used. Furthermore, the plasma concentrations of the antibiotics that were used were not assessed, which is information that could provide more insight regarding the association between treatment and outcomes. However, the findings of the study may form the basis for a larger investigation in the future.

## 4. Materials and Methods

This is a retrospective analysis. The study was conducted in a mixed ICU in the University Hospital of Larissa, Thessaly, Greece during a three-year period (2020–2022). Inclusion criteria were (a) admission to the ICU, (b) intubation and mechanical ventilation for >48 h, and (c) VAP caused by carbapenem resistant *K. pneumoniae* (CR-KP). Exclusion criteria were (a) age < 18 years old, (b) ICU readmission, (c) treatment with antibiotics other than CAZ–AVI + ATM or DCT, especially meropenem combined with ertapenem, for PDR strains, and (d) diagnosis on admission of COVID-19 infection. The first episode of VAP was accounted for. Patients were divided in three different groups: the first one consisted of patients that presented with VAP due to PDR *K. pneumoniae* and were treated with CAZ–AVI + ATM (CAZ–AVI + ATM group), the second consisted of patients with PDR *K. pneumoniae* and were treated with DCT (DCT group), and the third group consisted of patients that presented with VAP due to CR-KP and received appropriate therapy since the CR-KP strain was susceptible in vitro to at least to one antibiotic (control group).

### 4.1. Outcome

The primary outcome of the study was the change in the SOFA score and CPIS between baseline and the 4th day of antibiotic treatment. Secondary outcomes were the total duration of mechanical ventilation, SOFA score and CPIS evolution during the treatment period, and ICU and hospital mortality.

### 4.2. Antibiotics

CAZ–AVI and ATM were administered three times per day at the doses of 2.5 g and 2 g, respectively. Ertapenem’s dose was 1 g once per day, followed by meropenem at a dose of 2 g every 8 h. All of them were administered as extended infusions. For the control group, colistin was administered with a loading dose of 9M IU followed by 4.5 million IU twice per day; tigecycline as 100 mg twice daily after a loading dose of 200 mg; and finally, gentamicin 3–5 mg/kg as a single daily dose. All doses were adjusted for creatine clearance.

### 4.3. Clinical Assessment

For all study patients, the following characteristics were recorded: age, sex, characteristics of the respiratory system, illness severity based on the Acute Physiology and Chronic Health Evaluation Score II (APACHE II), SOFA score, CPIS, medical history, history of antibiotic use, type and duration of antibiotics used, and finally, the relevant clinical and laboratory findings. The patients’ treatment, including antibiotics, was left to the attending physicians’ discretion.

### 4.4. Microbiology

The Vitek 2 automated system (bioMerieux, Marcy l’ Etoile, France) was used for the identification and susceptibility testing of the isolated pathogens, combined with the disk diffusion method. For the interpretation of the results, EUCAST breakpoints were used.

### 4.5. Statistical Analysis

Categorical variables are presented as frequency (%), and they were compared using the chi-square test or Fisher’s exact test where appropriate. Continuous variables are presented as means (standard error, SE); they were compared using the Mann-Whitney *U* test, or Kruskal-Wallis test. The normality of data distribution was assessed using the Kolmogorov / Smirnov test. The Wilcoxon test was used to compare the variations in variables among the same group. Only variables with a *p* value < 0.05 were used in stepwise logistic regression models. ΔCPIS day 4 and ΔSOFA score day 4 were calculated as the difference between the value of the CPIS or SOFA score on day 4 and day 0. The same calculation was performed for ΔCPIS day 7 and ΔSOFA score day 7. GraphPad Prism 5 (GraphPad by Dotmatics Software Development, San Diego, CA, USA, 2007) software was used for data analysis.

## 5. Conclusions

Our data suggests that treatment with CAZ–AVI + ATM or DCT may offer clinical improvement for patients suffering from infections caused by PDR *K. pneumoniae* strains. It is an important finding since there are no antibiotics with in vitro susceptibility to treat such kinds of infections. However, larger studies are required to confirm the findings.

## Figures and Tables

**Table 1 antibiotics-13-00141-t001:** Baseline characteristics.

	CAZ–AVI + ATM Group (N = 12)	DCT Group (N = 11)	Control Group (N = 27)	All Patients (N = 50)
Age	54.2 (3.8)	66.4 (2.5)	54.1 (3.9)	54.2 (2.3)
Sex (male)	9 (75)	10 (91)	14 (52)	33 (66)
APACHE II score	14.8 (1.7)	16.5 (3)	14.3 (1.3)	15 (1.1)
SOFA score	7.3 (1.1)	7.3 (1.1)	8.2 (0.7)	7.3 (0.5)
Medical cause of admission	4 (33)	3 (27)	11 (40)	18 (36)
Presence of ARDS	5 (42)	5 (45)	7 (26)	17 (34)
PaO_2_/FiO_2_ ratio	168 (21)	184 (52)	255 (26)	216 (20) *
CPIS	4.1 (0.4)	3 (0.5)	3.7 (0.5)	3.6 (0.3)
WBC	13.183 (1.360)	12.817 (1.728)	14.036 (1.880)	12.888 (1.135)
CRP	4.4 (1.2)	6.2 (2.3)	11.2 (2.7)	8.4 (1.5)
Comorbidities				
COPD-Asthma	0 (0)	1 (9.1)	6 (22.2)	7 (14)
Cirrhosis	4 (33.3)	0 (0)	0 (0)	4 (8)
Heart Failure	2 (16.7)	2 (18.2)	1 (3.7)	5 (10)
Renal Failure	2 (16.7)	2 (18.2)	0 (0)	4 (8)
Immunosuppression	0 (0)	4 (36.4)	1 (3.7)	5 (10)

Data are presented as mean (SE) or N (%); CAZ-AVI, ceftazidime-avibactam; ATM, aztreonam; DCT, double carbapenem therapy; APACHE, Acute Physiology and Chronic Health Evaluation; SOFA, Sequential Organ Failure Assessment; CPIS, Clinical Pulmonary Infection Score; ARDS, acute respiratory distress syndrome; WBC, white blood cell; CRP, C-reactive protein; COPD, chronic obstructive pulmonary disease; *, *p* < 0.05 between the groups.

**Table 2 antibiotics-13-00141-t002:** SOFA score for day 0, day 4, and day 7 of VAP.

	CAZ–AVI + ATM (N = 12)	DCT (N = 11)	Control Group (N = 27)	*p*
Day 0	5.7 (0.7)	6 (0.8)	8.5 (0.7)	0.02
Day 4	4.7 (0.6)	5.5 (0.9)	7.3 (0.7) *	0.047
Day 7	4.5 (0.6)	4.7 (0.9)	7.2 (0.9)	0.07

Data are presented as mean (SE) or N (%); CAZ-AVI, ceftazidime-avibactam; ATM, aztreonam; DCT, double carbapenem therapy; SOFA, Sequential Organ Failure Assessment; VAP, ventilator associated pneumonia; *p*, comparison between the three groups; *, *p* < 0.05 between day 4 and day 0 of VAP for control group. Results by univariate analysis.

**Table 3 antibiotics-13-00141-t003:** CPIS score for day 0, day 4, and day 7 of VAP.

	CAZ–AVI + ATM (N = 12)	DCT (N = 11)	Control Group (N = 27)	*p*
Day 0	6.3 (0.5)	5.6 (0.5)	7.3 (0.3)	0.02
Day 4	5.7 (0.5)	4.9 (0.6)	5.7 (0.4) *	0.5
Day 7	5.5 (0.4)	4.6 (0.2)	5.6 (0.4) ^	0.06

Data are presented as mean (SE) or N (%); CAZ-AVI, ceftazidime-avibactam; ATM, aztreonam; DCT, double carbapenem therapy; CPIS, Clinical Pulmonary Infection Score; VAP, ventilator associated pneumonia; *p*, comparison between the three groups. *, *p* < 0.05 between day 4 and day 0 of VAP for control group; ^, *p* < 0.05 between day 7 and day 0 of VAP for control group Results by univariate analysis.

**Table 4 antibiotics-13-00141-t004:** Duration of MV, ICU stay, and mortality.

	CAZ–AVI + ATM (N = 12)	DCT (N = 11)	Control Group (N = 27)	*p*
MV duration (days)	70.7(25.3)	44.4(6.3)	36.4(9.3) *	0.02
ICU stay duration (days)	75(33.4)	56.5(8.4)	39.1(9.7) ^	0.03
Mortality (%)	58.3	33.3	37.5	0.4

Data are presented as mean (SE) or N (%); CAZ-AVI, ceftazidime-avibactam; ATM, aztreonam; DCT, double carbapenem therapy; MV, mechanical ventilation; ICU, intensive care unit; *, *p* < 0.05 between the control group and both the CAZ–AVI + ATM and DCT group; ^, *p* < 0.05 between the control and DCT group; *p*, comparison between the three groups. Results by univariate analysis.

## Data Availability

Data are available in UH Larissa database.

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
