# Peer review of "Meropenem plus Ertapenem and Ceftazidime–Avibactam plus Aztreonam for the Treatment of Ventilator Associated Pneumonia Caused by Pan-Drug Resistant Klebsiella pneumonia"

_antibiotics, 2024, doi:10.3390/antibiotics13020141_

Round 1

Reviewer 1 Report

Comments and Suggestions for Authors

My comments

I think it's a good study and the results are very interesting, because treatments for infections caused by bacteria that are multi-resistant to carbapenems are difficult to treat.

I just have a few points of clarification:

- The definition part should be in the introduction and the references for each definition should be given;

- in the methodology:

- The agreement of the ethics committee or authorisations from the Directorate-General for Health, or the informed consents of the patients included in the study, are missing.

- You should give the exact number of patients who were finally included in this study, and also give their frequency.

- line 118: why are you still excluding patients after selecting them according to your inclusion criteria? If your inclusion criteria are unreliable, they should be reworded.

I think this research would be more sound if antibiotic susceptibility tests were carried out after the various treatments to confirm the effectiveness of the treatments. For example, the different carbapenem molecules used could be tested using the diffusion disc method on Muller Hinton agar, followed by the Hodge test.

In my opinion, this is what's missing from this interesting study.

Reviewer 2 Report

Comments and Suggestions for Authors

Mantzarlis et al present an interesting analysis of antibiotic therapy in patients with Ventilator-associated pneumonia caused by pan-drug Klebsiella pneumoniae.

During a three-year period, the authors included 23 patients. Twelve were treated with a combination of ceftazidime-avibactam and aztreonam; the other 11 had a combination of two carbapenems.

The authors also evaluated 27 patients with carbapenem-resistant K pneumoniae but sensitive to at least one of the antibiotics used. 

Clinical evaluation was performed with SOFA and CPIS scores, along with time on invasive mechanical ventilation, ICU length of stay, and mortality. Scores trends were similar between the two groups with pan-drug resistant bacteria and the “control” group. ICU LOS and IMV LOS were lower in the “control” group, and higher in patients treated with ceftazidime-avibactam and aztreonam. Mortality was also higher in this last group.

This is an interesting exploratory study, reporting experience with antibiotic treatment of pan-drug-resistant K pneumoniae VAP.

However, I think that the manuscript lacks significant data that I hope the authors can provide.

1- First it is not clear if the patients received the different antibiotics as empirical therapy or as directed (rescue) therapy. It is important to know how much time elapsed from the diagnosis of VAP to the beginning of this therapy.

2- Second, the relationship between VAP and prognosis is not straightforward and several studies have shown a poor correlation. Consequently, it is important to know how the patients at the time of the diagnosis of VAP were: the authors provide the CPIS and SOFA. However, information about previous time on the ventilator and in the ICU, causes of admission, oxygenation ratio, and presence of ARDS, are also important. This is especially important as we are talking of a population with a very prolonged ICU stay (as stated in Table 6). 

3- Why did the authors choose the CPIS score? This has proven to be a poor marker of VAP or prognosis (eg, Clin Infect Dis, 2010: Suppl 1:S131 (doi: 10.1086/653062)). Remarkably, the authors reported a low mean value for CPIS (3.6).

4- The authors do not report the mechanisms of resistance that were found in the different populations. This has implications for the different antibiotic’s activity.

5- Also, the authors do not provide the list of active antibiotics in the control group. Especially, differences in the outcome of the different antibiotics, and if there were one or two active antibiotics, should also be informative.

Information regarding previous exposure to antibiotic therapy should be provided.

6- In results the authors repeated in the main text the results they presented on tables. This should be simplified.

7- How did the authors calculate the delta for SOFA and CPIS? The formula should be provided instead of presenting two different Tables.

8- According to Table 6, the group with the worst outcomes, by far, was the one treated with ceftazidime-avibactam plus aztreonam. Although I believe this was mainly a result of the different populations being treated, this differs from what is said in the abstract and conclusions.

9- In line 191, the authors state that treated patients can affect the course of the infection alleviating signs and symptoms, but no data is provided.

10- Also, in line 225, the authors state that in their study DCT therapy also had better clinical outcomes. Better than what?

Reviewer 3 Report

Comments and Suggestions for Authors

The paper deals with the evaluation of clinical efficacy of a double carbapenem regimen in alterative to other antibiotics regimens: 1)ceftazidime-avibactam plus aztreonam, 2) an antibiotic selected by in vitro susceptibility of pathological microrganism.

The study is very intersting as it apparently gets to ht econclusion that the double carbapenem regimen is a viable and better alternative to the other two regimens. However, the conclusion is obtained by a quite limited set of data. 

Of particular concern to us was the lack of quantitative doses for the antibiotics used. This is a strong limitation of the study since antibiotic resistance is strongly related to the concentration of active principles that depends on the dose, as well as on many other pharmacokinetics parameters that have not been considered in this study.

Another strong limitation is the lack of whatsoever description of the so called "appropriate therapy". The sentence "appropriate therapy was presented more beneficial" is therefore very difficult to understand.

A quantitative description also of the appropriate therapy was missing.

Discussion and conclusions of different clinical results must be supported by a deeper framing of the therapies used before these interesting clinical observations can be published in a reference journal such as Antibiotics.   

Comments on the Quality of English Language

English is essentially clear despite some criptic phrases such as "appropriate therapy.."

Round 2

Reviewer 2 Report

Comments and Suggestions for Authors

I am afraid tables 3 and 5, besides not providing any new information, have wrong calculations (for instance, table 3, control group day 7 should be -1,3 and not -0,86).

I suggest that the authors delete both of these tables. Calculations, if they think are important, may be referred to in the manuscript.

I do not have any other comments to add.

Reviewer 3 Report

Comments and Suggestions for Authors

The paper has been considerably improved by implementing most of the suggestions required. Limitations have been clearly discolsed.

Comments on the Quality of English Language

Minor flaws that could be fixed during copyediting
